# Propagation Capacity of Phage Display Peptide Libraries Is Affected by the Length and Conformation of Displayed Peptide

**DOI:** 10.3390/molecules28145318

**Published:** 2023-07-10

**Authors:** Danna Kamstrup Sell, Anders Wilgaard Sinkjaer, Babak Bakhshinejad, Andreas Kjaer

**Affiliations:** Department of Clinical Physiology and Nuclear Medicine & Cluster for Molecular Imaging, Copenhagen University Hospital—Rigshospitalet & Department of Biomedical Sciences, University of Copenhagen, 2200 Copenhagen, Denmark; danna@sund.ku.dk (D.K.S.);

**Keywords:** amplification, biopanning, commercial libraries, competitive propagation, cyclic conformation, enrichment factor, length, next-generation sequencing, peptide, phage display

## Abstract

The larger size and diversity of phage display peptide libraries enhance the probability of finding clinically valuable ligands. A simple way of increasing the throughput of selection is to mix multiple peptide libraries with different characteristics of displayed peptides and use it as biopanning input. In phage display, the peptide is genetically coupled with a biological entity (the phage), and the representation of peptides in the selection system is dependent on the propagation capacity of phages. Little is known about how the characteristics of displayed peptides affect the propagation capacity of the pooled library. In this work, next-generation sequencing (NGS) was used to investigate the amplification capacity of three widely used commercial phage display peptide libraries (Ph.D.™-7, Ph.D.™-12, and Ph.D.™-C7C from New England Biolabs). The three libraries were pooled and subjected to competitive propagation, and the proportion of each library in the pool was quantitated at two time points during propagation. The results of the inter-library competitive propagation assay led to the conclusion that the propagation capacity of phage libraries on a population level is decreased with increasing length and cyclic conformation of displayed peptides. Moreover, the enrichment factor (EF) analysis of the phage population revealed a higher propagation capacity of the Ph.D.^TM^-7 library. Our findings provide evidence for the contribution of the length and structural conformation of displayed peptides to the unequal propagation rates of phage display libraries and suggest that it is important to take peptide characteristics into account once pooling multiple combinatorial libraries for phage display selection through biopanning.

## 1. Introduction

The identification of molecules that bind to specific cellular targets is a critical step in the development of both diagnostics and therapeutics. Compared to large molecules such as antibodies, peptides offer many advantages for in vivo translation, including convenient synthesis, low production cost, effective tissue penetration, low immunogenicity, and amenability to chemical modifications [1,2,3,4]. Over the recent decades, phage display libraries have attracted broad attention as a means to discover high-affinity peptides that bind to a variety of biological targets [5,6,7,8,9]. In phage display selection, also known as biopanning, libraries of random peptides displayed on the surface of phage virions are presented to the target of interest. Subsequently, non-binding phages are removed, and phages with target-binding peptides are eluted and propagated in the host bacterium. High-affinity binders are enriched as the number of biopanning rounds is increased. Finally, the phage clones remaining in the eluted pool are analyzed by DNA sequencing to identify the sequence of target-binding peptides [10].

Up to now, a number of commercial and custom-made phage display peptide libraries have been constructed and reported in the literature [11,12,13,14,15,16,17]. These libraries are characterized by a variety of peptide structures in terms of length and conformation. The length of displayed peptides typically spans from 6 to 43 amino acids. Additionally, libraries expressing cyclic peptides have raised growing interest over the past years. Here, loop-conformation can be induced in different ways, such as the formation of disulfide bonds and chemical scaffolding [18,19]. Cyclic peptides exhibit several advantages, such as increased biological activity, structural rigidity, target selectivity, and biochemical stability compared to linear ones [20].

Even though biopanning enables high throughput screening of large peptide libraries, it is difficult to predict in advance which type of library will lead to the identification of the most promising binders for a specific target. In general, the success of phage display selection to identify target binders heavily relies on library diversity [15]. The use of more diverse libraries increases the probability of finding peptides with better binding properties. The large size of the library is thus desired to select peptides that not only bind to challenging targets but also possess additional properties such as high stability or the ability to bind to multiple related targets [21,22]. In line with this, some attempts have been made to increase the size of phage display libraries [16,21,23,24,25]. Although these studies have shown promise, technical hurdles pose challenges to the generation of very large phage display libraries. Another strategy to cover a broader sequence space and increase the throughput of selection is to mix multiple libraries with different peptide structures and use the pooled library as the input of biopanning [25,26,27,28,29,30,31]. To be able to use this strategy for biopanning, it is necessary to investigate the impact of propagation on the composition of pooled libraries. Phage display selections contain a propagation step in which phages are amplified in the host bacterium. For the selection to be only driven by binding affinity to the target, all phage clones should have an equal capacity for amplification. However, this is not an achievable scenario, and it has been uncovered that individual phage clones within a library possess an advantageous propagation capacity compared to others. Propagation capacity is determined by an increase in the copy number of phage clones over a given time period during propagation. The higher propagation capacity of individual clones can be attributed to different types of mutations that occur in the phage genome [27,32,33,34] or the lack of surface display of peptides (wild-type clones (WT) constituting bias in the library construction) [35]. The presence of phages with faster propagation rates causes a major problem for the library selection, leading to the enrichment of false positive hits—i.e., clones that do not have high binding affinity to the target but dominate the phage pool solely due to increased propagation capacity. In this work, we investigated the effect of displayed peptide length and conformation on the population-level propagation capacity of phage display peptide libraries in a competitive environment. This was done by mixing three widely used commercially available phage display peptide libraries from New England Biolabs (Ph.D.^TM^-7, Ph.D.^TM^-12, and Ph.D.^TM^-C7C) and subjecting the mixed pool to propagation. These libraries display two different peptide conformations: cyclic and linear. Ph.D.^TM^-7 and Ph.D.^TM^-12 are linear libraries with peptides of 7 and 12 amino acids, respectively. The Ph.D.^TM^-C7C library displays seven amino acid peptides flanked by a pair of cysteine residues, which form disulfide bonds during virus assembly, and the peptide is finally displayed as a loop on the phage [36]. We used next-generation sequencing (NGS) to determine the propagation capacity of the phage libraries. Our data provided support for the notion that the propagation capacity at the population level is affected by the length and structural conformation of displayed peptides in a way that the capacity is decreased with increasing peptide length and even more with cyclic conformation. The findings of the current study can be used to better understand the impact of peptide characteristics on the propagation rate of phage pools during biopanning, in particular, once a mixture of multiple combinatorial libraries is used as the input for biopanning.

## 2. Results

In-house MATLAB^®^ and Python scripts were used to analyze NGS raw data and conduct sequence filtering. As indicated in Appendix A, filtering of Illumina sequencing data (based on the criteria defined in Section 4.4) resulted in the removal of about 16–21% of the reads across all three time points, indicating an approximately similar percentage of removed reads for all time points. The cleaned reads were used for all the subsequent analyses.

### 2.1. The Percentage of Wild-Type Clones Remains Approximately Balanced during Competitive Propagation Assay

Sequences representing WT clones do not express the displayed peptide and GGGS linker and are, therefore, placed in removed reads. These clones may have superior infectivity and propagation capacity compared to phages with a surface-displayed peptide [37]. Since WT clones are present in the competitive propagation assay, they can compete with and affect the propagation capacity of the peptide-expressing clones [38]. Hence, we assessed the percentage of WT clones in the entire phage pool (by dividing the number of WT reads by the total number of reads at each time point). Our calculations revealed that WT clones constitute 12.5% of the phage pool at the beginning of the assay, and this percentage remains approximately balanced during the assay at 7.4% and 10.3% at 150 and 270 min, respectively. According to our results, the proportion of WT clones undergoes no substantial change throughout the competitive propagation assay. Also, we found that WT clones make up 60.3%, 47.3%, and 64.1% of the removed reads at time points 0, 150, and 270 min, respectively. Based on this, the total percentage of removed reads should not be taken as a representation of the quality of high-throughput sequencing data.

WT clones can arise from each library and are known by the sequence AETVESCLAKSH, which represents the sequence immediately downstream of the peptide insert at the N-terminus of the phage coat protein pIII. This sequence is the same for WT clones from all libraries, and therefore, it is not possible to distinguish which library the WT clones originate from. As the WT clones do not contain the linker sequence GGGS and are placed in removed reads, the MATLAB-based data sorting will result in the exclusion of these clones from the subsequent analyses.

### 2.2. Increased Length and Cyclic Conformation of Displayed Peptides Reduce the Propagation Capacity of Phage Display Peptide Libraries in a Competitive Environment

To investigate the relative propagation capacity of different libraries in a competitive environment, we amplified a phage pool containing 10^9^ pfu of each of the three NEB libraries in an early log culture of ER2738 bacterial strain for 270 min. At three time points (0, 150 min, and 270 min), samples were withdrawn from the culture, their DNA was extracted, and the purified DNA was analyzed by NGS. The analysis of the NGS-based competitive propagation assay revealed that the two linear libraries Ph.D.-7 and Ph.D.-12 exhibited a considerably larger increase in the number of peptide-expressing clones compared to the cyclic Ph.D.-C7C library during the propagation period (Figure 1A).

Even though the same pfu was used as input to the assay and no amplification occurred at time point zero, we observed a difference in the number of phages among the three libraries at this time point. The phage input was calculated based on the titer stated by NEB and only subjected to 100-fold dilution prior to the assay. To minimize the impact of this bias on the conclusions of the competitive propagation assay, we calculated the fold change in the number of peptide-expressing phages from different libraries from the start to the end of the amplification time (Figure 1B). Our NGS data indicated that the order of propagation capacity in an inter-library competitive environment is Ph.D.-7 > Ph.D.-12 > Ph.D.-C7C. This suggests that in the pool of phage libraries, the population of phages expressing cyclic peptides is more hindered in propagation compared to a phage population expressing linear peptides and that a greater length of displayed peptides (by comparing the two linear libraries) reduces the propagation capacity of a phage population.

### 2.3. Diversity of the Phage Pool Is Reduced during Competitive Propagation

To assess the entire phage pool diversity (all three libraries), we calculated the percentage of distinct (singletons) and repeated sequences across all three time points (Figure 2). The percentage of distinct sequences for each library was calculated by counting the number of distinct sequences from each library and dividing by the total number of sequences from all three libraries. The percentage of distinct sequences indicates the diversity of the phage pool. We observed a decrease in overall phage pool diversity with increasing propagation time (Figure 2). Before amplification (t = 0 min), distinct sequences represented 96% of the entire phage pool. At time points 150 and 270 min, the percentage of distinct sequences decreased to 87% and 49%, respectively. Additionally, we assessed the contribution of distinct sequences from each of the three libraries to the whole phage pool. This enabled us to determine which libraries underwent the biggest and smallest loss in diversity during competitive propagation. We found that the largest loss of diversity was for the Ph.D.-12 library, and the smallest loss of diversity was for the Ph.D.-7 library. Also, the greatest loss in diversity was observed in the latter part of the assay (150–270 min) compared to the initial part.

### 2.4. Enrichment Factor Analysis Indicates That the Ph.D.-7 Library Has the Highest Propagation Capacity in the Competitive Environment

The enrichment factor describes fold change in the frequency (copy number) of a specific phage clone between two consecutive time points during propagation. Typically, absolute frequency (copy number of each clone) is used to evaluate the enrichment of peptides displayed by a clone during biopanning and propagation. In the context of propagation, a higher frequency of a peptide would suggest a higher propagation capacity of the clone displaying that peptide. Enrichment based on absolute frequency cannot reflect the true enrichment of clones during propagation because the total number of phage-derived sequences differs between time points. To gain a more rigorous evaluation of the propagation capacity, relative enrichment should be utilized, which is determined by comparing the relative frequencies of a sequence at two consecutive (earlier and later) time points. To this end, we calculated the enrichment factor of all sequences which overlapped between two consecutive time points (0–150 min and 150–270 min) and counted the number of sequences from each library in the EF-sorted dataset. A sequence with EF > 1 represents an increased copy number (enrichment) for the phage clone displaying that sequence during propagation, and thus, we only included sequences with EF > 1 in our analysis (Table 1). In this manner, we have excluded phage clones that do not show enrichment, and their copy number has been reduced during the propagation period. Therefore, we can have a more exact assessment of the contribution of each library to the total propagation capacity of the phage pool.

As indicated in Table 1, the proportion of overlapping clones belonging to the Ph.D.-12 library dominated when comparing 0–150 min enrichment. However, moving forward through propagation, there is a substantial reduction in the number of sequences from the Ph.D.-12 library and a minor reduction in the number of sequences from the Ph.D.-C7C library. Additionally, this is accompanied by a remarkable increase in the number of sequences from the Ph.D.-7 library, which provides evidence for the higher propagation capacity of this library in the competitive environment. 

**Table 1 molecules-28-05318-t001:** The percentage of sequences from each library with EF > 1 between consecutive time points during propagation (t = 0–150 min and 150–270 min).

	0 to 150 min	150 to 270 min
Ph.D.-7	9.0%	51.7%
Ph.D.-12	69.4%	34.7%
Ph.D.-C7C	21.6%	13.6%

By illustrating the distribution of library-specific EFs across all three libraries at time points 0–150 min and 150–270 min (Figure 3), we see that the higher contribution of sequences from the Ph.D.-7 library at 150 to 270 min is represented by multiple sequences with a broad range of EFs. This reflects that the higher contribution of Ph.D.-7 sequences to the second half of propagation (150 to 270 min) is not simply due to a few very fast-propagating Ph.D.-7-originated clones but rather a large number of Ph.D.-7 clones with heterogenous EFs. This details that the seemingly higher propagation capacity of the Ph.D.-7 library from 150 to 270 min is related to the ability of numerous Ph.D.-7 clones to propagate at a high frequency (EF > 1).

### 2.5. Propagation Capacity of Phage Display Peptide Libraries Differs in Competitive and Non-Competitive Environment

We investigated the absolute propagation capacity of libraries in a non-competitive manner by amplifying each library in a separate bacterial culture and determining titer through the plaque count method (Figure 4).

In this assay, we sought to have the same input titer of phages (10^9^ pfu) for each of the three libraries. However, it is evident that the input titer of the Ph.D.-12 library was significantly higher compared to Ph.D.-7 and Ph.D.-C7C, presumably resulting from imprecise reported titers from the manufacturer as well as sampling bias. At t = 150 min and t = 270 min, the highest titer increase was observed for Ph.D.-C7C. The results of this experiment presented a contrast to the competitive propagation assay, highlighting that even though Ph.D.-C7C library has the highest absolute propagation capacity among the three libraries, it showed the lowest propagation capacity in a competitive context.

## 3. Discussion

As phage display peptide libraries remain widely used in the identification of clinically relevant ligands, continuous efforts have been devoted to increasing the diversity of selection input to improve the probability of identifying promising peptide ligands through biopanning [15]. Pooling peptide libraries (i.e., by mixing libraries with different peptide lengths and conformations) is an easy and accessible way of obtaining an increased size and diversity of the selection input [25,26,27,28,29,30,31]. This enables the discovery of peptides with target binding abilities amongst billions of peptide variants. The binding affinity selection of peptides is coupled with a propagation step which leads to an increased copy number of enriched sequences to enable the detection of selected peptides. However, the downside of propagation is the undesirable enrichment of fast-propagating clones with target-unrelated peptides [32,39]. There are a limited number of studies in the literature on how propagation can impact the composition of a phage display peptide library and lead to a biased selection of target-binding peptides in biopanning. In addition, there is far less information about the comparison of the propagation capacity of libraries with different lengths and conformations of displayed peptides. There is only one report investigating the effect of the pIII display of peptides with different lengths and conformations on phage viability [13]. They found a 10-fold decrease in plaque formation for clones expressing peptides of 43 amino acids with a central cysteine compared to clones expressing peptides of 37 amino acids. As clones from each of the two libraries can contain multiple and varying numbers of cysteines, the cyclic conformation of clones within one library differs (peptides can be linear, cyclic, or even bicyclic) [13]. However, these results are based on Sanger sequencing and the plaque count method, which makes it difficult to conclude the true effect of peptide structure on phage propagation. Additionally, studies where libraries of different lengths and conformations have been pooled for biopanning do not refer to the impact of propagation on the composition of the pooled library. The vast majority of these studies are based on Sanger sequencing, which provides a limited sequence space of the target-recovered phage pool, and there are few reports using NGS to analyze the selection output of pooled libraries. To our knowledge, the current work is the first use of NGS to compare the propagation capacity of different libraries (different lengths and conformations) in a competitive environment.

In the M13-based phage display system of Ph.D.-12, -7, and -C7C libraries, the displayed peptide is expressed as a fusion to the N-terminus of the phage minor coat protein pIII. This protein is involved in phage adsorption to the host bacterium and, thereby, plays an important role in the initial step of phage life cycle [36,40,41,42]. The expression of an exogenous peptide could put the fusion phage at a selective disadvantage compared to insertless phages and affect the phage capacity to propagate [36,37]. However, an unavoidable artefact of library construction is the presence of phage clones without expressed peptides (WT clones). Due to this artefact, we calculated the percentage of WT clones throughout the course of amplification in the competitive propagation assay. The absence of WT-associated corruption of the phage pool during propagation (seen as a balanced WT percentage throughout the assay) confirms that the assay measures competitive propagation between the peptide libraries rather than between libraries and WT clones. Furthermore, we found that the majority of removed reads are WT clones, and the percentage of WT clones correlates with the percentage of removed reads at all time points (Appendix A). The WT clones could originate from any of the three libraries, and the WT sequences were therefore excluded from the analysis of the contribution of each library to the phage pool. Obviously, this entails some bias to the analysis since the WT clones were present in the assay and competed for infecting bacteria in both the competitive and non-competitive assays. In this sense, the competitive assay is advantageous compared to non-competitive one as we were able to identify and exclude WT clones from the analysis in a competitive assay, but it was not possible to assess the contribution of WT clones to the non-competitive assay.

In the competitive propagation assay, we showed that the library propagation capacity decreased with increasing peptide length and cyclic conformation (Figure 1). The NGS analysis uncovered a bias before the initiation of propagation in which Ph.D.-12 constituted almost half (48.8%) of the phage pool (Figure 1A). This increased proportion of clones from the Ph.D.-12 library could originate from several possible factors; initial bias in the titer of the three libraries, sampling bias from the library to the assay medium, sampling bias from the assay pool, and bias in the steps of NGS sample preparation (DNA extraction from phage virions, PCR amplification, purification of PCR products, sequencing, and NGS analysis where sequences with errors are removed). Due to this bias, the analysis was based on fold change (Figure 1B) of library proportion from time point t = 0 to t = 150 and from t = 0 to t = 270 min. However, despite the initial bias resulting from the higher copies of virions from the Ph.D.-12 library, the Ph.D.-7 library clones revealed the largest fold increase compared to the two other libraries.

We assessed the diversity loss for the entire phage pool and for each library by calculating the percentage of distinct sequences (sequences with frequency = 1) and the number of repeated sequences (sequences with frequency > 1) (Figure 2). Diversity loss during phage pool propagation is expected and even desirable when caused by target binding (removal of non-binding sequences) in biopanning [38]. However, this diversity loss can also be due to the propagation competition that typically follows the elution of target-bound phages. The diversity loss of the individual libraries in the competitive propagation assay exemplifies that length and cyclic conformation affect the extent of diversity loss and, thereby, the loss of potential hits inter-library. As such, the Ph.D.-7 library seemingly has the least diversity loss compared to Ph.D.-C7C and Ph.D.-12.

Calculation of the relative enrichment factor for all the clones that overlapped between two consecutive time points (sequences with EF > 1) can provide a more exact analysis of the propagation capacity of the three libraries in the competitive context. This analysis is advantageous for two reasons. First, it overcomes any bias arising from differences in the initial number of virions from each library, which would cause the competition to be unequal, by calculating the enrichment of sequences based on the relative frequencies of each sequence in two consecutive time points. Secondly, it allows us to investigate propagation capacity by analyzing a particular pool of phage clones showing enrichment during the propagation period (the best propagators within each library). This is highly important as clones with EF < 1 whose relative frequencies were decreased during propagation and did not show enrichment were excluded from the analysis. EF analysis interestingly showed that from the start of propagation to 150 min, the majority (69.4%) of clones arose from Ph.D.-12 (Table 1). However, from 150 to 270 min, the majority (51.7%) of clones were from the Ph.D.-7 library, which reveals that this library outcompeted the other libraries in the second half of the assay. This finding suggests that even though clones from the Ph.D.-7 library were not the most abundant at 0 min and 150 min, their propagation capacity outcompeted the other libraries with the progress of propagation. The fact that the proportion of Ph.D.-7 expanded from 9% to 57.1 % reveals that clones from this library possess a greater propagation capacity compared to the percentage of both Ph.D.-12 and Ph.D.-C7C, which declined in the second half of the assay (Table 1). The higher propagation capacity of the Ph.D.-7 library compared to other libraries is also backed by analyzing the distribution of EF values (Figure 3). EF distribution exhibits a higher number of virions from the Ph.D.-12 library (denoted as red dots) in the first half of propagation (Figure 3A, t = 0–150 min), which results from the high number of the members of this library at the beginning of the assay. However, the number of virions from the Ph.D.-7 library (denoted as blue dots) is highly increased in the second half of propagation (Figure 3B, t = 150–270 min). In the second half of propagation, we also see how the distribution of EF values of the Ph.D.-7 sequences are spread out and very heterogeneous compared to the first half of propagation. This indicates that the high propagation capacity of the Ph.D.-7 library is the result of numerous clones rather than a limited number. Based on the results of EF analysis, we can conclude that the Ph.D.-7 library possesses the greatest propagation capacity in the competitive environment. Even though Ph.D.-7 is the fastest library, some clones from the other two libraries with high EF values can also be found in the EF-sorted dataset (Appendix A), suggesting their significant enrichment in the pool. These clones, and especially the ones from Ph.D.-C7C, are interesting as they may represent fast-propagating clones displaying propagation-related target-unrelated peptides (Pr-TUPs). Additionally, these clones exemplify heterogeneity within each library, highlighting that even though Ph.D.-C7C is considered the slowest library, it might still contain some clones with high propagation capacity. 

In the competitive assay, we already investigated the relative propagation capacity of the three phage display libraries. We conducted a non-competitive assay to evaluate the absolute propagation rate of each library (Figure 4). Here, libraries were cultured separately and titered at time points matching that of the competitive propagation assay. When comparing the fold changes in titer over time, the highest propagation rate was seen for Ph.D.-C7C. The contrast in results between competitive and non-competitive environments could exemplify the population-based differences in the propagation capacity of phage libraries. Even though Ph.D.-C7C propagates faster than the two other libraries when cultured isolated, it is outcompeted by libraries of linear composition when cultured in the pool of all libraries. This can provide further evidence for the intra-library heterogeneity and the presence of some fast-propagating clones in the Ph. D.-C7 library. The observed contrast could also likely be explained by the different methodologies applied in the two assays. In the non-competitive assay, which is based on the plaque count method, only infective phage particles are counted, whereas, in the competitive assay, DNA is counted irrespective of being derived from an infective or non-infective phage particle. When considering the effect of peptide length and conformation on propagation, we believe that the competitive assay is more accurate as it enables propagation assessment in an inter-library manner. Additionally, by basing the competitive propagation assay on NGS analysis, we obtain insights into the identity of the clones and the diversity of the pool.

The efficiency of the selection of clones with optimal binding can be hampered by an unequal representation of different phage clones and alteration of their composition that arises during phage propagation. This phenomenon has been well documented on a clonal level where one clone (expressing a particular peptide) undergoes genetic changes that render it a more efficient propagator compared to other clones of the library [27]. Additionally, a previous report [43] has identified only linear peptides from five rounds of biopanning using a mixture of linear and disulfide-bridged cyclic libraries in a T7 phage display system. Since our results indicate that mixing structurally different libraries will favor the propagation of phages expressing linear peptides over cyclic ones, it is obvious to consider whether the findings reported in [43] are due to the target binding motif only being present on the linear peptides or if the phages expressing cyclic peptides have simply lost the propagation race.

## 4. Materials and Methods

### 4.1. Phage Display Peptide Libraries

The three commercial phage display peptide libraries Ph.D.^TM^-7 (Lot number:10043452), Ph.D.^TM^-12 (Lot number: 10057682), and Ph.D.^TM^-C7C (Lot number:10043274) from New England Biolabs (Ipswich, MA, USA) were used in our work. These libraries are characterized by a display of random peptide sequences in different formats as fusions to the N-terminus of the M13KE phage minor coat protein pIII. A tetra amino acid linker sequence (Gly-Gly-Gly-Ser) exists between the displayed peptide and the mature pIII of the phage. The Ph.D.-7 and Ph.D.-12 libraries display linear hepta- and dodecapeptides in the form of A-X7-GGGS and A-X12-GGGS, respectively (X being any canonical amino acid). These two libraries were provided as stocks with the reported titer of 2 × 10^13^ pfu/mL. The Ph.D.-C7C library displayed heptamer peptides with a cysteine-based disulfide-constrained loop in the form of AC-X7-CGGGS and was provided with the reported titer of 1 × 10^13^ pfu/mL. The libraries were purchased collectively and treated in the same manner prior to and throughout the experiments.

### 4.2. NGS-Based Competitive Propagation Assay

An overnight culture of the Escherichia coli strain ER2738 was diluted 1:100 with 100 mL of LB medium. Simultaneously, 1 × 10^9^ pfu of each library was added to the early log bacterial culture, and the flask was quickly whirled and incubated at 37 °C for 270 min with vigorous shaking (250 rpm). At time points 0, 150, and 270 min, 3 mL of the culture was withdrawn for DNA extraction. To recover amplified phages, the samples were centrifuged (4000× *g*) at 4 °C for 10 min to remove bacterial cells and debris. Phages were purified by incubating with a precipitation buffer (1/6 volume of 20% PEG/2.5M NaCl) at 4 °C overnight and centrifugation (12,000× *g*, 4 °C, 10 min). The purified phages were stored at 4 °C until DNA extraction.

### 4.3. NGS Sample Preparation

Phage DNA was extracted as previously described (34) using a NucleoSpin ^®^ Plasmid kit for the isolation of M13 DNA (Macherey-Nagel, Düren, Germany). DNA was eluted in 20 µL of RNAase-free water and subjected to PCR using Q5 High-Fidelity 2X Master Mix (New England Biolabs, Ipswich, MA, USA). Reaction volumes were 50 µL comprised of 25 µL of master mix, 5 µL of phage DNA, and 2.5 µL of each of the (10 µM) forward- and reverse primers (10 µM). The primers consisted of phage genome-specific sequences and Illumina-compatible sequencing adapters:

Forward: 5′-AATGATACGGCGACCACCGAGATCTACACTTCCTTTAGTGGTACCTTTCTATTCTC*A

Reverse: 5′-CAAGCAGAAGACGGCATACGAGATCGGTCTATGGGATTTTGCTAAACAACTTT*C

PCR cyclic conditions were initial denaturation at 98 °C for 30 s followed by 20 cycles of denaturation at 98 °C for 10 s, annealing at 60 °C for 30 s, and extension at 72 °C for 20 s with final extension at 72 °C for 2 min. The PCR product was purified using a QIAquick PCR purification kit (Qiagen, Düren, Germany), and the purified DNA was subjected to NGS analysis through a single-end sequencing strategy using the MiSeq platform and the MiSeq v2 Nano reagent kit. NGS was conducted by the Center for Genomic Medicine, Copenhagen University Hospital, Rigshospitalet (Copenhagen, Denmark).

### 4.4. NGS Data Analysis

NGS raw data provided as FASTQ files were analyzed using a MATLAB^®^ script (Appendix A) which converted nucleotides into amino acids and filtered peptide sequences into cleaned and removed reads. Cleaned reads were characterized by lacking invalid amino acids (‘*’) within the variable (displayed peptide-encoding) region and bearing the linker sequence (GGGS) after the variable region in the phage genome. In Ph.D-C7C library, the presence of cysteine residues (AC- and C) flanking the variable region was also used to identify the cleaned reads.

Due to the complexity of analyzing sequencing data of three different Ph.D. libraries in one pool, three different Python scripts were developed and applied to NGS datasets to eliminate these complications.

Python script 1 (Appendix A): Each of the MATLAB script-generated removed reads also contained cleaned reads belonging to the other libraries (e.g., Ph.D.-7 and -12 cleaned reads being placed in the Ph.D.-C7C removed reads as they do not contain the cysteine at positions 2 and 10). To solve this problem, script 1 was used to identify and remove these duplicate sequences from the removed reads using the cleaned reads as the reference. Finally, the three library-specific removed reads were pooled into one group of removed reads for each time point. This was done for MATLAB-generated removed reads for all libraries and all time points.

Python script 2 (Appendix A): The MATLAB script-generated cleaned reads of the Ph.D.-7 library also included several Ph.D.-12 library-specific cleaned reads (when a Ph.D.-12 AA sequence contains the GGGS motif at both positions 8–11 and 13–16). These duplicates were removed from the Ph.D.-7 library and kept in the Ph.D.-12 sorted library via a second Python script. This was done for all MATLAB-generated cleaned reads of the two libraries and all time points.

Python script 3 (Appendix A): The last script was developed to identify the number of insertless (WT) clones, which do not display any peptide. This was achieved by identifying the number of reads containing the sequence AETVESCLAKSH and its variants (differing one amino acid from the above sequence). The reads with one amino difference from an abundant sequence most likely derive from sequencing errors. The sequence AETVESCLAKSH represents the N-terminus of the mature pIII of the M13 phage. This Python script was used for all removed reads from all time points. The Python scripts were developed on Spyder and are available in Appendix A.

### 4.5. Calculation of Enrichment Factor

To gain a better insight into the propagation capacity of each library in the competitive environment, the enrichment factor (EF) was calculated for all sequences overlapping at two time points (sequences common between time points 0 and 150 and sequences common between time points 150 and 270). EF was calculated by dividing the relative frequency of a given sequence at a certain time point by the relative frequency of the sequence at an earlier time point. The relative frequency of each sequence was already calculated by dividing the absolute frequency of the sequence by the total number of reads (for all libraries) at each time point. All sequences were finally sorted based on EF (from the highest to lowest). EF can show how many folds the frequency of a sequence has changed during the propagation period. As EF < 1 represents de-enrichment (reduced copy number) during propagation, we included only sequences with EF above 1 in our analysis.

### 4.6. Non-Competitive Propagation Assay

Each library was amplified separately by simultaneously adding 1 × 10^9^ pfu to an ER2738 culture (OD600 = 0.1 to have the same multiplicity of infection as the competitive propagation assay). All three libraries were amplified at 37 °C and 250 rpm with favorable aeration. The 1 mL samples were withdrawn at time points 0, 150, and 270 min and stored at 4 °C until titration. Samples were diluted (10-fold) in duplicates, and 10 µL of diluted phage was incubated with 200 µL of ER2738 culture (OD600 = 0.5) for 5 min before being resuspended in 3 mL of top agar (0.7%) and plated onto LB/Tet/X-gal/IPTG plates. Plates were incubated at 37 °C overnight, and blue plaques were counted the following day to determine the phage titer.

## 5. Conclusions

Our data provide evidence for the notion that unequal representation in phage selections happens on a population level when libraries of different lengths and structural conformation of the displayed peptide are pooled. It should be kept in mind that even though mixing peptide libraries of different conformations increases the diversity of selection input (which increases the likelihood of identifying promising peptide hits), an unequal representation of the slowest library will affect the selection process if the population propagation capacity is not taken into account. This work is the first to compare the propagation capacity of different libraries in a competitive environment using NGS analysis. Our findings lead us to the conclusion that the propagation of pooled libraries with different structures of displayed peptides can result in a biased composition of the phage pool, favoring the selection of phages with short and linear peptides during biopanning.

## Figures and Tables

**Figure 1 molecules-28-05318-f001:**
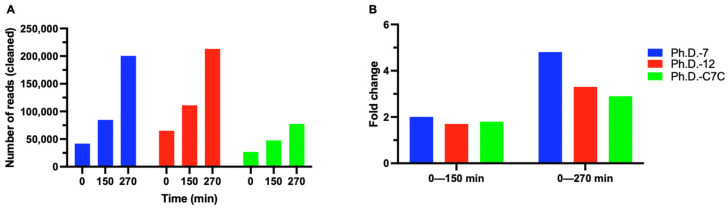
NGS-based inter-library competitive propagation assay. Based on NGS data analysis, all peptide sequences were categorized according to the library origin and counted. (**A**) The number of cleaned reads is depicted for each library during the propagation period. (**B**) To overcome the bias caused by unequal numbers of phage particles from different libraries, the fold change of phage numbers was calculated for each library during the propagation period.

**Figure 2 molecules-28-05318-f002:**
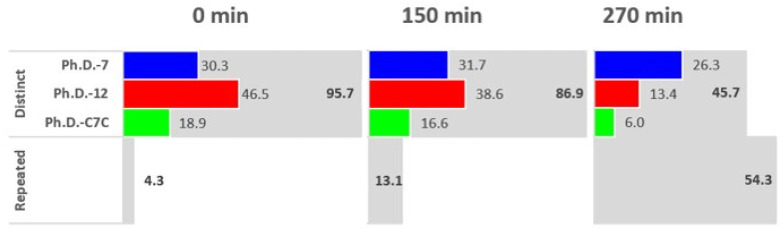
Phage pool diversity. Bar chart showing the percentage of distinct and repeated sequences and the percentage of each library within distinct sequences. The number of repeated sequences, which was used for the calculation of percentage, was obtained by excluding one copy and including the remaining copies of each repeated sequence.

**Figure 3 molecules-28-05318-f003:**
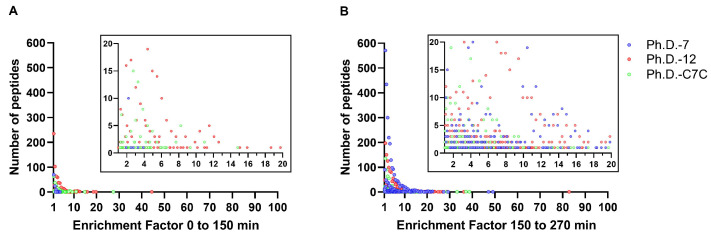
Dot plot of enrichment factor distribution from 0 to 150 min (**A**) and from 150 to 270 min (**B**). Each dot represents the number of peptides/sequences (*y*-axis) with the corresponding enrichment factor (*x*-axis) for each library.

**Figure 4 molecules-28-05318-f004:**
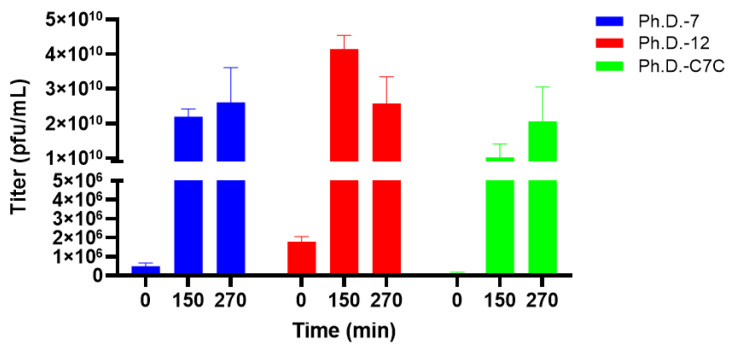
Non-competitive propagation of phage display peptide libraries based on plaque count method. From each library, 10^9^ pfu were cultured separately, and the titer was determined by the plaque count method (SEM error bars).

## Data Availability

Data are contained within the article or Appendix A. Data that support the findings of this study are available from the corresponding author upon reasonable request.

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
