# Peer review of "Propagation Capacity of Phage Display Peptide Libraries Is Affected by the Length and Conformation of Displayed Peptide"

_molecules, 2023, doi:10.3390/molecules28145318_

Round 1

Reviewer 1 Report

This paper presents a promising approach to using next-generation sequencing (NGS) to investigate the amplification capacity of three widely used commercial phage display peptide libraries. I appreciate the innovative nature of your work and would like to provide some constructive feedback to further strengthen your findings.

1, It would be better to rewrite your abstract, please use common style and academic language, delete “background” “Method” “Results” “” Conclusions”.

2, Please double-check the reference, I notice many mistakes in that part, please correct the abbreviation of the journals.

3, Can you elaborate on the difference between peptide libraries in the main text?

Reviewer 2 Report

This is a potentially interesting analysis of the behavior of phage displayed peptide libraries under competition. Although the authors focus on investigating the differences in propagation of three distinct libraries when pooled, there is extensive evidence that within each of these libraries different peptide sequences have different rates of propogation (as they discuss in lines 70-80).  Therefore it is important for the authors to state that their conclusions are relevant to the overall (average?) properties of each library and that specific sequences within each library will generate a breadth of propogation rates such that there will be significant overlap among the libraries.  

  The authors could improve the manuscript significantly by defining, up front, a number of terms that are used extensively throughout. These include:

propagation capacity of phage virions

enrichment factor

Specific points for clarification:

The authors state 'filtering of Illumina sequencing data resulted in the removal of about 16-21% of the reads across all three time points' - what are the criteria for 'filtering' ?  Presumably 'cleaned reads' are the reads that were not filtered out?  The term is not defined.  

The authors state 'Since WT clones are present in the competitive propagation assay and might affect the propagation ability of the peptide-expressing clones' - By what mechanism might the presence of WT clones affect the 'propagation ability' of other clones? Can the authors define 'propogation ability'?  Is this the same as 'propogation capacity'?

Figure 1: Why are the number of cleaned reads so different for the three libraries when 10**9 of each library were pooled.  Shouldn't they be the same?  Or was the quality of reads that different for the three libraries?  'fold change' was used because of bias in the different number of phage for each library.  But the number was explicitly fixed as being the same...  what am I missing?  These facts are mentioned in lines 137-140, but no reason for them is given. Additional discussion on lines 210-217 is relevant, but the reader could easily miss them.

Line 143 - how was propogation capacity calculated? (i.e.  data used and the equations that derive propogation capacity from the data)

The authors make broadly generalized statements on the basis of three libraries: 'population of phages expressing cyclic peptides are more hindered in propagation compared to a phage population expressing linear peptides and that a greater length/size of displayed peptides (by comparing the two linear libraries) reduces the propagation capacity of a phage population.'  Maybe.  But maybe not proven?

The authors asses 'phage pool diversity' based on repeated sequences.  How is 'diversity' defined?  How is it calculated from the occurrence of repeated seqeuences?  

Figure 2 does not allow the reader to distinguish between a few repeated sequences that dominate the culture vs a broad range of sequences that occur only a few times each.  The functional diversity of the library may very well be quite different for those two possiblilities.  Reference to discussion on lines 189-196 would be relevant.

I find lines 177-179 confusing.  

Table 1: 'percentage of sequences from each library with EF > 1'  is this percentage calculated on the basis of percentage of distinct sequences?  Or is each sequence weighted by the number of occurrences?  in other words, in a library of 100 sequences with one sequence having EF>1 occuring 99 times and one sequence with EF<1 occurring once, is the percentage of sequences with EF>1 99% or 50%?  

Final conclusion: 'We have shown that pooling libraries with different structures of displayed peptide can lead to a biased selection, favoring phages with short and linear peptides.'  At least in these cases.  Does this make an argument for NOT pooling libraries in biopanning experiments?  

some minor awkward sentence structures.  could be cleaned up, but not really necessary/

Round 2

Reviewer 1 Report

The revised manuscript looks much better, thanks for author's effort.